# Intelligent Process Automation: An Application in Manufacturing Industry

Federico A. Lievano-Martínez [1], Javier D. Fernández-Ledesma [1], Daniel Burgos [2,3], John W. Branch-Bedoya [3] and Jovani A. Jimenez-Builes [3,*]

[1] Departamento de Ingeniería, Universidad Pontificia Bolivariana, Medellín 050030, Antioquia, Colombia; federico.lievano@upb.edu.co (F.A.L.-M.); javier.fernandez@upb.edu.co (J.D.F.-L.)
[2] Instituto de Investigación, Innovación y Tecnología Educativas (UNIR iTED), Universidad Internacional de La Rioja, 26006 Logroño, La Rioja, Spain; daniel.burgos@unir.net or daburgoss@unal.edu.co
[3] Departamento de Ciencias de la Computación y de la Decisión, Facultad de Minas, Universidad Nacional de Colombia, Medellín 050035, Antioquia, Colombia; jwbranch@unal.edu.co
\* Correspondence: jajimen1@unal.edu.co; Tel.:+57-4255-222

**Abstract:** Background: The intelligent processes automation has been cataloged as one of the most potential and strategic technology solutions to develop a corporate digital transformation. Method: This paper introduces essential concepts to create Intelligent Process Automation (IPA) in industries and proposes a framework to implement IPA technologies successfully. The approach involves: firstly, assembling a good implementation setup and deeply researching the process using process mining techniques. Secondly, choosing and locating the best AI technology inside the IPA. Finally, defining an appropriate architecture of the IPA. Results: The paper illustrates an IPA use case in the manufacturing industry, where it is possible to automate the process of sending production orders to a manufacturing plant and optimize waste and plant capacity significantly. Conclusions: The research depicts the potential of intelligent process automation and its quantifiable benefits in the manufacturing process, and the contribution can be applied to different enterprises with a global context.

**Keywords:** cognitive automation; intelligent process automation; robotic process automation; artificial intelligence





## 1. Introduction

Companies are increasingly looking for fundamental digital transformation where systems can become intelligent to adapt to constantly evolving business dynamics and sensitive buyers' preferences [1,2]. Furthermore, customers continually demand business systems be adaptive and responsive enough not only to carry out their daily workload but also to provide additional insights to them to be able to make knowledgeable and more trustworthy decisions [3–5].

As the global economy changes in response to the development of new technologies, businesses must become more agile and respond quickly to requirements, goals, and challenging customer demands [6]. Moreover, competitive and financial pressures force organizations to be more efficient, pushing them to look for new technologies and methodologies that could help them be more productive than competitors, save costs, and add value to their businesses [7].

Robotic Process Automation (RPA) is a highly novel intelligent solution for implementing digital transformation in business processes. RPA can improve human work, especially service responsibilities and management activities within organizations. RPA has recently been introduced into human resources, finance, accounting, supply chain management, and Information Technologies (IT) systems due to the increased pressure to improve service and operational efficiencies.

Multiple studies have revealed the many benefits of RPA implementations within an organization (e.g., [2,6,8]). However, RPA is more powerful when supported by artificial intelligence (AI) and cognitive automation creating the new IPA technology. There are fundamental differences between RPA and IPA, such as the type of task faced. RPA face Routine task, and IPA face task related to judgments and decision making. RPA follows instructions, IPA takes decisions and gives conclusions. The scope of RPA is broader. IPA needs to be focused on a specific goal. The cost and time to implement IPA are higher than RPA, and IPA is an emergent technology.

Therefore, developing and implementing intelligent RPA technologies with AI-supportive platforms makes business processes more intelligent through either full automation without human intervention or partial human intervention [9,10].

RPA and AI are commonly considered separate approaches in the literature [3]. However, by combining both, expansive innovation possibilities and transformations can resolve challenges created by voluminous judgment-related tasks and enhance decision making within organizations. Forrester [10] estimated that by 2021, over four million RPAs will automate repeatable tasks, but the focus will shift towards AI integrations and RPA analytics enhancements. Similarly, Everest Group explained that most buyers were delighted with RPA solutions, but they required analytics and cognitive capabilities. Therefore, the primary purpose of this article is to lay out a new and novel technology that walks through AI incorporation into robotic process automation.

It should be noted that the novelty of this article is: first, an attractive research work is carried out to define and clarify the concept of intelligent process automation consulted academic and business documents. Second, a framework is proposed to implement IPA in industries not identified in the literature to date, generating a working roadmap for companies that wish to develop this type of solution. Third, the potential of artificial intelligence combined with RPA to solve complex problems that cannot be solved independently is identified. Finally, a practical exercise in a manufacturing industry that other industries can replicate under global contexts is exposed.

The paper is distributed as follows: in section two, the basic concepts of RPA are presented. Then, in section three, the concepts of IPA are externalized. Then, in section four, a methodology is shown, which will be implemented in section five (Framework to Implement IPA in Industries). In section six, a case is presented showing application in the manufacturing industry. Finally, conclusions and references are indicated.

## 2. The Basic of Robotic Process Automation

RPA's primary purpose is to mimic human work and then substitute it with RPA [4]. Robotic process automation is a type of software that imitates daily human interactions with computers. This means that software might replace processes that were previously done by humans, such as logging into a system, entering data, executing workflows, and any rule-based activity done in a computer, especially interactions with CRMs such as SalesForce or Hubspot, information systems such as SAP, email, and shared drives, among others. RPAs replicate human activities without human intervention. This software is called a robot because it is auto driven. RPAs can read emails, open attachments, move files, follow programmed rules, extract data, integrate information with ERPs, CRMs, and HR systems, and much more.

Their applications are numerous: insurance, capital markets, banking, finance and accounting, business administration, and many more [5]. This type of automation helps the employees focus on more critical tasks, be more innovative, and dedicate time to enhance their knowledge and skillsets. Hence, RPA technology demonstrates increasing profitability and throughput when recurrent and repeatable processes are automated. An RPA approach has certain features that should be considered:

- RPAs do not require an entirely new software system to be developed; in contrast, it assumes that IT systems are already developed and running.
- These systems should not be replaced or changed; instead, they should be manipulated by the RPA.
- RPAs aim to produce software programs that run through the user interface of other computer systems in the way a human would use the system.

The popularity of RPA owes to its increased speed and resources available to run processes, which makes it highly efficient compared to the human workforce. For example, these RPAs are cheaper, do not demand overtime allowances, and work 24 h and seven days per week. Additionally, they offer higher accuracy, compliance, and speed. The added value of RPA is mainly related to organizational performance improvement and cost reduction by increasing workforce capacity with digital and low-cost RPAs to run routine business processes and shift full-time employees (FTE) to perform value-added activities, improving work quality [3].

However, not everything is positive about RPA. It also has weak points that open the door to new paradigms, methodologies, and use cases such as Intelligent Process Automation (IPA). Some of those are:

- Detailed and accurate process mining is mandatory for successful RPA implementation. Relevant activities to be automated must be clarified to generate actionable business insights.
- The input–output rules must be clearly defined regarding synchronization between dependencies and external systems involved. RPAs do not know how to handle scenarios that have not been mapped before. New cases or scenarios would entail changes in the robot's logic, rules, or inputs.
- Voluminous activities should be accompanied by rigorous technological infrastructure. RPAs are excellent at applying rule-based reasoning, but they are limited to enhancing decision making in complex processes involving a higher abstraction level.
- RPAs are not suitable when working with judgment-related tasks, and AI addition must be considered.

The significant need for good process mining, the inability to adapt to changing conditions, the lack of stability related to data volumes, the inconsistency in decision making, and the impossibility of dealing with judgment-related tasks certainly unlock the door for Intelligent Process Automation (IPA) to step in.

## 3. Intelligent Process Automation

RPAs can be called IPAs when endowed with AI capabilities that help overcome the weaknesses above. This is achieved by creating built-in AI capabilities such as machine learning, deep learning, natural language processing, other algorithms, and APIs (Application Programming Interfaces) that help with forecasting, analyzing, and optimizing data to solve problems [11].

The IEEE Standards Association defines Intelligent Process Automation (IPA) as a *"preconfigured software instance that combines business rules, experience-based context determination logic, and decision criteria to initiate and execute multiple interrelated human and automated processes in a dynamic context"* [12].

Intelligent Process Automation differs from conventional automation in its ability to imitate human activities and reproduce human decision making at crucial stages. For example, when traditional rule-based automation is used with added cognitive capabilities, a new kind of RPA emerges. RPA replicates rational judgment and intellectual skills and leverages computational capabilities to deliver beyond human capacities [8]. An IPA extends traditional RPA functionalities with new emerging technologies, such as self-learning capabilities, data mining process discovery, optimization models, AI-screen recognition, speech algorithms, image recognition, machine learning, and natural language processing [13].

RPA tools enhance their functionality with the objectives of AI being extended with the use of artificial neural network algorithms, text mining techniques, and natural language processing techniques for the extraction of information and the consequent optimization and forecasting to improve the operational and business processes of organizations [11]. In this context, IPA plays an important role when:

- The input data source is unstructured, and it is thus necessary to use NLP algorithms to extract the information and carry out the task efficiently.
- The RPA cannot adapt to changing conditions.
- The RPA cannot deal with any data.
- The RPA is unable to turn data into insights.
- The RPA is unable to enhance decision-making- and judgment-related tasks.

In this way, when combining RPA automation capabilities with AI skills, an enterprise can improve its operations and customers' interactions and optimize efficiency, gain deep process insights, and most importantly, create new business models (see Table 1). In addition, many companies are beginning to recognize the potential benefits of combining these two technologies to overcome challenges, especially when operation efficiencies become too complex to understand and therefore to improve, leading to uncertain and inaccurate decision making. These types of complexities need to be addressed with appropriate AI algorithms and mathematical procedures if real insights are too be found [11,13].

**Table 1.** RPA vs. IPA (adapted from [11]).

| Feature | Simple RPA | Cognitive RPA or IPA |
|---|---|---|
| Type of tasks | Routine tasks | Non-routine tasks |
| Capability of robot | Follow instructions | Come to conclusions |
| Application focus | Broader | Narrower |
| Market offerings | Maturing | Emerging |
| Implementation and ongoing costs | Lower | Higher |
| Implementation timeframe | Weeks | Months |

Machine learning, for instance, can develop prediction capabilities using the data that is being processed by the RPA and provide forecasts with a certain accuracy level, which will increase as the learning process improves, enabling companies to be prepared and eventually able to control upcoming events. Additionally, deep learning and neural convolutional network capabilities can be used to solve problems such as pattern classification, data interpretation, and new optimal possibilities discovery. Another option might be to enhance RPAs with natural language processing algorithms that standardize and extract relevant data from documents, emails, audio, videos, and non-structured inputs, adapting to any possible condition and reducing the need for human interaction [10].

Intelligent process automation is not just one tool. It can be thought of as consisting of six core technologies: RPA, the main one, combined with machine learning, natural language processing, artificial vision, deep learning, and mathematical programming [13–15]. However, when combining the automation capabilities of RPA with AI, care must be taken with the methodology and process to successfully implement the project. Hence, several points of concern must be kept in mind: the importance of process mining, the core AI technology utilized to resolve the problem, the robust design of the system, and the computing architecture used. Furthermore, it is crucial to consider servers consumption, real-time backend-frontend efficiency, velocity, and the need to be focused on the problem solution, always aiming to improve the most critical enterprise KPI (Key Performance Indicator) [8,10].

## 4. Methodology

### 4.1. The Implementation of Digital Technologies in the Industries

Technical innovation is the main economic development engine that helps companies adapt at an organizational level to market dynamics dictated by globalization [16]. Mar-

kets, professions, and society face rapid and radical change due to the maturation and penetration of digital technologies (DT) in most industry markets and domains. Digital transformation generates changes at different levels:

- Process (reduction of manual tasks).
- Organization (new ways of doing things and new services).
- Business (changing roles in the value chain).
- Society (changing social structures).

Countries in the most advanced stage of digitization obtain 20% more economic benefits than those in the initial stage. Although the importance of digitization is well known, companies often struggle to understand the potential impact and benefits of implementing DT [17,18].

A global business study developed by MIT revealed that 26% of companies are in the early stages of digitalization, 45% considered their company in the developing period, and 29% considered themselves mature companies in terms of digitalization. These last ones usually have a clear digital strategy combined with a collaborative leadership culture that has fueled transformation and encouraged risk taking. However, this study also identified a lack of digital transformation research, i.e., digital transformation management or cost identification, primarily due to limited empirical evidence and ambiguity on formulating and reevaluating DT implementation [6,17].

Companies with a coherent plan to integrate physical and digital components in operations will most likely be successful in their business model's transformation towards a digital mechanism and therefore towards technological innovation [19]. However, recent studies indicate that although digital transformation is a well-known idea and one of the most common discussions among entrepreneurs, there is no method or clear and organized plan for DT implementation that enables business model transformation [19]. The digital transformation phenomenon has been extensively explored in different academic domains, which has produced a damaged overview of implementation and practice [20,21]. However, few studies clearly define digital transformation implementations within business models, nor do they offer a well-defined methodology that contains instructions on implementing DT [22].

The implementation of DT seems to depend more on a methodological point of view than on a theoretical definition, and this argument is justified by the fact that DT implementation is seen as one of the natural and most considerable difficulties in all industries undergoing digitalization [20,23]. Without exception, companies face numerous obstacles that prevent them from implementing DT [16,23]. Therefore, a deep understanding of successfully carrying out these transformations and applying DTs in the business community is still necessary [18].

*4.2. Intelligent Process Automation in the Literature and Consulting Websites*

Over the last decade, there has been steady progress towards RPA. However, we are currently at an inflection point in its evolution towards IPA. This new paradigm integrates machine learning and AI into the classic RPA.

IPA research focuses on non-routine tasks, jobs requiring human cognition by learning from experience, and unstructured data processing. However, the applications for IPA are still emerging.

The purpose of this section is to identify studies around this theme. Nevertheless, as it is a highly novel topic, few studies and research can be found in the literature. Table 2 shows relevant studies and enterprise documents related to IPA. In the research, the following research questions were formulated to proceed with the systematic review of the literature:

P1. Is it possible to overcome the problems of technological adoption using good practices to develop RPA and IPA?

Yes, RPA and IPA technologies intrinsically bring adoption issues because they are emergent technologies, and the implementation concepts are already being discovered. However, studying the literature and business cases (Table 2) and proposing good practices

based on experience and successful industry implementations helps overcome the problems and contribute to knowledge adoption.

P2. What factors enable a company's stable and profitable technology adoption when using RPA-IPA? What is the influence of determinants on business processes using RPA-IPA?

In this research, the experimental exercises applied in industries showed representative factors that enable the profitable implementation of RPA and IPA. They are the experience, the profitability, and the complexity, which are resolved with adequate methods such as process mining, the proposal architecture, and the artificial intelligence algorithm chosen to resolve the problem.

P3. Is it possible to understand the RPA technology adoption phenomenon by experimenting with real organizations and experts in the field?

The RPA and IPA adoption can be understood through experimentation in the current organization, as is shown in chapter 6. The actual use cases allow an understanding of the factors that affect technology adoption and propose theoretical frameworks to develop successful RPA and IPA projects in industries.

P4. Is it possible to improve the RPA-IPA performance metrics?

This research argues that using the proposed theoretical framework and approaches defined in the below sections can improve the performance in the implementation of RPA-IPA projects and increase the benefits in automation and process metrics.

The research is limit to the following specialized databases: Science Direct, Springer Journals, IEEE Xplore, Scimago Journal, Scopus and searchers such as Google Scholar articles published after 2018. We filtered the search to thematic areas of computer science and engineering, with this query "TITLE-ABS-KEY (robotic process automation AND intelligent process automation AND cognitive automation AND digital robots)".

**Table 2.** Relevant IPA studies and consulting documents.

| Authors | Topic | Implementation |
|---|---|---|
| [14] | The main contribution of this paper is providing a review of AI and RPA contributions to Industry 4.0. | No applied cases are presented. |
| [24] | Study how recent advances in machine intelligence are disrupting the world of business processes. | No applied cases are presented. |
| [25] | Present a framework that combines RPAs with conversational agents (or chatbots) to create an interactive business process automation solution. | The Bank customer is submitting a loan application, and IPA is processing this request to determine approbation or rejection. |
| [26] | Provide building blocks using Microsoft Azure for intelligent or cognitive robotics process automation. However, the integrations with RPA are not exposed. | Five applications are presented: extract intent from audio, email classification and triage for automated support-ticket generation, a case of fraudulent credit card transactions, cross-correlation in time series problem, and understanding traffic patterns. |
| [27] | This article considers the potential for cognitive algorithms to disrupt knowledge work in the modern workplace. | No applied cases are presented. |
| [28] | Framework to implement IPA in business. | Two business cases are presented: A communications company using IPA to respond to customer queries and resolve service issues; and an event ticketing company using IPA to automate one of their most important weekly tasks related to queries to the database. |
| [29] | Framework to implement IPA in business. | Two business cases are presented: IPA to reduce the need for manual work in US grocery delivery and a second solution for an RCM (revenue cycle management) service provider's complete eligibility and benefits verification process. |

## 5. Framework to Implement IPA in Industries

The framework for implementing IPA in industries is presented in Figure 1 and then, in the following subsections, each of the components is described.

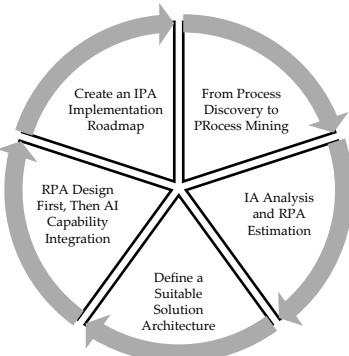

**Figure 1.** Framework to implement IPA in industries.

### 5.1. Create an IPA Implementation Roadmap

An IPA roadmap should consider the end-to-end process instead of simple tasks. It needs to have a holistic picture to show the added value obtained when a simple RPA is not enough and where the power of AI is needed to gain more benefits and deliver increased business outcomes. Process mining is key to determining the project design and accurately deciding the technology to implement [11].

On the other hand, governance is fundamental. Therefore, the segregation of responsibilities and mechanisms review must ensure conflicts are swiftly dealt with, such as a decision matrix framework, programed meetings, and reporting rules [30,31].

Additionally, it is imperative to run a cost–benefit analysis. IPA implementations can involve complex integrations with APIs and costly AI platforms, which can cause the solutions to be very expensive, making it impossible for the company to reach a ROI (return of investment). Therefore, IT must keep in mind that simple solutions with fast development life cycles are always better. The added value is given by the creativity and innovative use of the data [15].

It is also necessary to track benefits, KPIs, and measures that cover different aspects of the intelligent automation solution, such as process performance indicators and productivity metrics [8]. Finally, it is crucial to understand that the impact of intelligent automation goes beyond automating tasks and the use of the data that they produce. The extent to which data can be used in algorithms, techniques, and data science technologies influences the success of implementing IPA.

Finally, changes in management impact functions, roles, and teams in this type of implementation. For example, roadmaps should include a timetable and different items that need to be considered to adequately address the underlying impacts of technology from the organizational, human, and operational perspectives. The intelligent automation journey requires meticulous planning, coordinated action, and a great deal of rigor to be successful [14,15].

### 5.2. From Process Discovery to Process Mining

Process blueprints are typically formed by surveying process owners and subject matter experts, conducting workshops with relevant people, and observing the environment and employees' different jobs [9,32,33]. It is evident that defining a process model implies a lot of structuring work, e.g., relevant business activities and events must be identified and named, the order of these activities must be well defined, and business logic must be specified within an organized and structured scheme. However, this discovery process can also be automated and oriented by a new paradigm called "Process Mining", which involves simulation, AI, self-learning algorithms, and data science to extract knowledge and process structure.

Process mining comes from data mining data generated while executing business applications to design and extract the correct logic that depicts the business. Process mining uses data to automatically build an as-is process model and improve the efficiency and speed in this critical step of the automation [32,33].

Because of this, it provides a holistic view of processes and identifies bottlenecks, variations, and underlying root causes of inefficiencies, enabling engineers to improve their efficiency quickly and identify where to implement RPA-IPA. Through this, automation can be implemented 54% faster, and its value increases by 44% when combined with process mining. When developing an IPA, process mining is essential because it:

- Is key to understanding the current state of the processes and identifying inefficiencies and optimizations that can be covered using IA.
- Streamlines the identification of parts of the process where IA could be crucial.
- Identifies where IA should be implemented.
- Monitors automation rates, process compliance, and other KPIs.
- Can be very costly if IPA is implemented without sufficient understanding of your business processes.

We can question whether the process models obtained by process mining are equal to hand-crafted models regarding their quality. Process mining can only be applied when appropriate data is available and may omit steps that are not data driven. If process discovery is supported by data and is unaffected by employees' judgments, it may be a good candidate for process mining and possibly will be competent for developing a successful IPA.

### 5.3. AI Analysis and RPA Estimation

During process mining and RPA project estimation, the AI capability integrated into the robot should be analyzed. It is essential to identify the phase of the project and the specific process step in which AI would be applied. It is good to follow these recommendations:

- Prioritize stable applications and high ROI projects (investment return) for RPA and AI implementation.
- Measure insights and KPIs that justify and support AI implementation.
- Define the cost–benefit of implementing AI platforms; sometimes, the implementation is more expensive than profit.
- Be sure to have enough good quality data to run algorithms, especially if you use self-learning algorithms or machine learning applications.
- Be sure to have IT resources that ensure sustainable solutions and input–output transitions.
- Define an exemplary architecture of the solution that allows smooth deployment and robustness.
- Finally, try to have a robust solution focus. It is better to have a robust but straightforward AI implementation than complex algorithms that reduce the RPA's sustainability.

### 5.4. Define a Suitable Solution Architecture

The ability and experience to support a holistic architecture that covers the key components of intelligent automation is a fundamental factor in the success of implementing an IPA project. As a result, organizations need to evaluate capabilities in this area to ensure that new systems integrate seamlessly with the overall enterprise ecosystem. In addition, the IT area must establish a performance monitoring mechanism to manage and monitor the intelligent automation projects in their entirety.

It is an excellent practice to implement the AI component using cloud infrastructure; this allows scalability and interoperability to ensure good algorithm performance and integration with automation procedures.

Here are some suggestions to define a suitable architecture:

- The data is the core of the AI algorithms; for this reason, try to choose optimal databases to store and expose the data according to the nature (structured, semi-structured, or unstructured) of the data.
- It is possible to implement ETLs inside the core of the robots; this facilitates the integration with other data sources and AI and APIs.
- Store the inputs and outputs of the AI algorithms in efficient data repository technology (as far as possible columnar and using distributed computing). This practice drastically improves the performance of the IPAs.
- If a cloud architecture is used to develop the AI component, it is vital to choose a cloud provider according to the IT scheme of the company. This method facilitates the integration and sustainability of the solution.
- Finally, making the IT area an integral part of projects is an exceptional decision. The IPAs must be co-created by the business and IT teams to ensure the success of the projects. Below in Figure 2, a typical architecture to develop IPAs under a holistic scheme is shown.

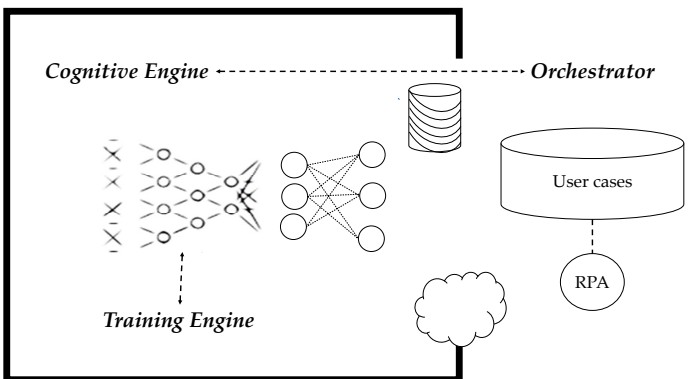

**Figure 2.** IPA common architecture example.

*5.5. RPA Design First, Then AI Capability Integration*

It is preferable to start with RPA and then add the AI capability. There is no simple way to develop an IPA, but it is essential to consider that connected, knowledgeable, and organized data is necessary to apply cognitive and AI techniques. The RPA technology helps map processes' inputs and outputs, allowing data maturity for AI implementations.

Choosing the first set of processes where AI can be applied is critical for the entire program to succeed. These must be processes where success is almost ensured or processes known as "low-hanging fruits", where inconveniences, exceptions, management changes, and transition efforts are minimal. Once the initial integrations succeed, then the AI capabilities can be expanded [10,34].

The most common IPA applications are those in which algorithms play a vital role in identifying data patterns and decision making observed in the process flow. Scenarios that involve prediction technologies and planning algorithms are also options that an IPA might face. On the other hand, scenarios with large bottlenecks, cognitive analysis, and optimization logic are good candidates for IPA [8].

However, there are other situations where human validation is required to move the process along. In such cases, a machine learning algorithm can provide recommendations regarding the best set of possible actions in the process to a human employee. For example, IPA can use many tools, such as natural language processing, deep learning, and computer vision to evolve the RPAs. The following section shows an actual use case application that has been built using the methodology outlined above.

## 6. Application in the Manufacturing Industry

In this chapter, an application case is presented in the manufacturing industry where the automation of sending production orders to a fabric finishing plant is required. However, the immense potential of developing artificial intelligence on this automation (RPA) was identified, incorporating an AI algorithm that would allow optimization in the allocation and cutting of fabric rolls as the primary raw material of this process. The chapter is developed following the conceptual framework set out above and describes the results and benefits of developing this IPA for this type of manufacturing process. The sequence of steps is presented in Figure 3 and then, in the following subsections, each of them is described.

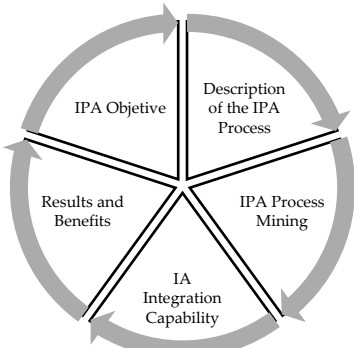

**Figure 3.** Sequence of instructions to apply the proposed technology in manufacturing industry.

### 6.1. IPA Objective

The IPA's objective is to automate the process of creating purchase orders for fabric stamping. The robot must receive the information in an email from a previously defined template, process it, create the purchase order in SAP and generate an email response with the order file in PDF format.

### 6.2. Description of the IPA Process

The process begins with an incoming customer email; the attached file contains a set of fabric purchasing requirements of different qualities, colors, and types. First, this information is validated and then sent to the orchestrator to activate the robot. If there are orders in the queue, the robot logs into SAP and creates the order by entering all the information from the orchestrator into an SAP transaction.

A significant challenge arises when business rules come in. First, there is an available pool of fabric rolls, and the robot must optimize the pickup of the fabric rolls to ensure fabric waste minimization and compliance with the requirements. Hence, an AI algorithm was integrated into the RPA and used to optimize the decision making and pick up the fabric rolls in less time than usual.

Different ways of integrating the AI algorithm with the RPA were explored. In the end, incorporating the code inside the robot's programming was the best option for this exercise. In this way, the robot had control of the exceptions and input and output interactions. After deciding the most optimal set of rolls in the program, minimizing fabric waste, and satisfying the requirement from the email request, the robot finally creates the purchase order in SAP. The process ends with making a PDF file, which is saved in a folder with specific naming conventions and is sent to the organization manager and supplier via email.

### 6.3. IPA Process Mining

Table 3 shows the description of the process mining carried out in the IPA.

**Table 3.** IPA Case—Process steps.

| Steps | Process Description of the IPA Case |
|:---:|---|
| 1 | The RPA software checks if there are unread emails in the mailbox. |
| 2 | If there are new unread emails, the request type is identified by the email subject. If the issue does not exist or is not within the standards, a notification email is sent back requesting the email information to be corrected. |
| 3 | After the subject is correctly identified, the RPA software verifies that the email attachment is in a xlsx format. |
| 4 | Mandatory parameters are verified. If the file contains invalid parameters, the RPA sends the information rows with problems back to the sender. |
| 5 | If all parameters contained in the file are correct, the requested orders are sorted and prioritized by date. |
| 6 | Key process information from the file is sent to the RPA orchestrator: material, delivery cycle, value, brand, quality, the quantity of fabric required, ID of the fabric supplier, minimum quantity to take, and priority of the order, among others. |
| 7 | If there are no unread emails, the RPA checks for pending orders in the orchestrator's queue. |
| 8 | The RPA logs into SAP. |
| 9 | The RPA searches for the transaction code. |
| 10 | Material numbers and order due dates are entered into the system. |
| 11 | The RPA verifies the existence of the required material in SAP. If it does not exist in SAP or the due date is a non-working day, then an email is sent informing of the error, and this task is marked as failed. |
| 12 | Value, required quantity, brand, and quality of the raw material are entered into the system. |
| 13 | The RPA verifies if the data was entered correctly. |
| 14 | Fabric supplier ID, order number, arrival warehouse, brand, and fabric quality information are entered. |
| 15 | Fabric batches are assigned according to business rules and restrictions, assuring that available fabric meets the minimum required quantity in the order. |
| 16 | The AI software is invoked to find the set of batches that together meet the amount of required fabric and minimize waste the most. |
| 17 | The output of the AI software is put into an Excel file that contains the information of the batches to be taken and the quantities for each one. |
| 18 | Specified fabric batches and quantities are selected in SAP. |
| 19 | The order is saved, and the order number is extracted. |
| 20 | The order is generated, and a PDF is saved. |
| 21 | The supplier email is extracted. |
| 22 | An email is sent to the supplier and buyer with the order and the generated PDF. |
| 23 | The RPA software checks if there are unread emails in the mailbox. |
| 24 | If there are new unread emails, the request type is identified by the email subject. If the issue does not exist or is not under the standards, a notification email is sent back, requesting email information to be corrected. |
| 25 | After the subject is correctly identified, the RPA software verifies that the email attachment is in a xlsx format. |
| 26 | Mandatory parameters are verified. If the file contains invalid parameters, the RPA sends the information rows with problems back to the sender. |
| 27 | If all parameters contained in the file are correct, the requested orders are sorted and prioritized by date. |
| 28 | Key process information from the file is sent to the RPA orchestrator: material, delivery cycle, value, brand, quality, the quantity of fabric required, ID of the fabric supplier, minimum quantity to take, and priority of the order, among others. |
| 29 | If there are no unread emails, the RPA checks for pending orders in the orchestrator's queue. |

### 6.4. AI Integration Capability

The AI component allows optimal batches to be found and delivers a result that satisfies the following business conditions:

- The rolls of the same batch should be consumed.
- The roll can be picked from its warehouse or an external supplier. If the case is an external supplier, rolls cannot be split.
- Rolls cannot be left with less than 50 m.

The following tolerances for the order are handled:

- If the fabric requirement ranges from 0 to 600 m, the order cannot exceed 10% additional fabric.

- If the fabric requirement ranges from 601 to 1000 m, the order cannot exceed 6% additional fabric.
- If the fabric requirement exceeds 1000 m, the order cannot exceed 2% of additional fabric.

An AI component programmed in a Python algorithm was implemented. After exploring different ways to perform the integration, a useful RPA built-in activity that converts the Python code to Net code was the best option. Hence, execution errors can be displayed and controlled inside the robot monitoring because the robot platform was built in Net code having overall control of the algorithm.

By integrating both the AI software and RPA with well-defined business rules, a decision-making capability is added to the automation, allowing an enhanced experience where software can make optimal decisions complying with business expectations as a human would. Table 4 shows a process breakdown from a technical perspective, specifying process steps performed by the RPA and AI software.

**Table 4.** IPA Case design.

| Steps | Explanation |
|---|---|
| 1 | The RPA software generates a query to the company's WMS system, which stores inventory information, and exports it to a SQL database, appending detailed requirement information to the data dump. |
| 2 | The RPA software calls the AI software using the server console where the system is housed. |
| 3 | The AI software reads the SQL database generated in step 1. |
| 4 | The AI software creates a global query, considering demand and inventory rolls available that contain over 30 m of fabric. If the inventory is more extensive than demand, the AI software starts execution. Otherwise, it stops and sends an error email to the RPA software console, finishing execution. |
| 5 | The AI software then sorts existing fabric categories in the different business units of the company. Each business unit stores a different fabric quality. To optimize the inventory and reduce waste, they need to be consumed in a specific order (First A, then B, and C). |
| 6 | The AI software takes a subset of the fabric of category A. |
| 7 | The AI software executes three possible scenarios, defined in the business requirements as rules to optimize waste and keep the fabric batches homogeneous. |
| 8 | First scenario: AI software finds the identical quantity within one single batch. A heuristic algorithm is applied to minimize the difference between a total requirement and the combination of rolls of the same batch that gets close to the requirement with a delta that ranges from −10 m to +50 m. If it does not find a solution for the first scenario, it moves to the second one. Second scenario: the AI software must decide how much fabric to take from each roll within every batch. In this case, the delta does not apply to the requirement established by the business. Therefore, the AI software needs to virtually "cut" a fabric amount from a roll of any batch. In this case, the AI algorithm decides the right amount and selects from multiple rolls the right one to minimize the fabric waste after the cutting process. For this scenario, the AI software needs to leave more than 50 m of fabric in the roll after the cut to be utilized in further requirements and not treated as waste. The amount of fabric (of a single category) required is not within a single roll, so the AI software needs to pick from different rolls until the requirement is met or the closest amount to the requirement is found, always keeping the business deltas. |
| 9 | The AI software generates a JSON file containing the execution message according to the scenario. |
| 10 | The AI software sends a message to the RPA software with the execution results through the RPA software console. |
| 11 | The RPA decides what to do next according to the message received. It can stop the execution and communicate errors or continue with reserving the fabric in the system by choosing the amount of fabric that the AI software indicated in the message. |

### 6.5. Results and Benefits

The IPA implementation obtained the following impacts:

- Freed up two employees' time.
- The process reduced the time spent on each order from 12–15 min to 5.1 min.
- The implementation of AI software reduces wasted fabric by 30%, generating economic benefits.
- The prioritization of the orders made by the IPA allowed them to meet urgent requirements and streamline the processing pipeline; see Table 5.

**Table 5.** IPA use case results and metrics.

| Requirements | Fabric Inventory Available | Fabric Selected by the IPA | Comments |
|---|---|---|---|
| 120 m Just one category type should be picked. | Ten different of fabric rolls with different categories are available. | The IPA chose 101.57 m of Batch 8000504453 18.43 m was cut of Batch 8000504452 | The IPA met the requirement with zero fabric waste. |
| 125 m Two types of categories should be picked. | Seven different fabric rolls with two types of categories are available. | The IPA selected in total 52.60 m of batch with code 8000504455 and 72.4 m of batch with code 8000504516. Here, it was not necessary to cut any roll. | Out of seven different fabric rolls, the automation picked the categories expected by the user according to the business rules. The selected fabric rolls minimize the waste to 5%. Before this automation, the average waste percentage was 12%. |
| 300 m Just one type of categories should be picked. | Five different rolls of the same categories are available. | The IPA chose the entire 102.9 m of Batch 8000504516 Additionally, 147 m of Batch 8000504522 50.1 m was cut of Batch 8000504529 | The IPA met the requirement with 0 fabric waste. |
| 465.38 m Just one type of categories should be picked. | 124 different rolls of the same categories are available. | The IPA chose the entire 75.18 m of Batch 8000501572 76.86 m of Batch 8000501574 77.7 m of Batch 8000501585 78.54 m of Batch 8000501575 78.54 m of Batch 8000501578 Additionally, 78.54 m of Batch 8000501583 | The absolute difference with the total required amount is: −0.02 m. |

## 7. Conclusions

RPA and AI are two very different technologies that complement each other very well. This paper shows how these technologies can be combined and a methodology for building and successfully deploying IPAs.

RPA combined with AI capabilities undoubtedly enhance their separate capabilities, offering an excellent alternative to resolve complex problems and, at the same time, provide the company with solutions to challenges that could not be addressed by RPA alone. These new scenarios open the door to new business possibilities and allow enterprises to discover insights that add value to decision making.

Intelligent process automation creates new ways of effecting digital transformation in an enterprise. The added value that is limited by the usual automation is surpassed by intelligent automation, bringing new self-learning and cognitive capabilities to the table and, therefore, new solutions to tough challenges never before assessed.

The determinants raised in the article for the execution of RPA projects incorporating AI are validated with an important use case in the manufacturing industry. In this way, the potential of this conceptual framework and the benefit of this mixture of technologies to solve highly complex industrial problems is demonstrated. In addition, the application case presented in the manufacturing industry opens the way to develop automation of complex issues that have not yet been resolved in the literature and the corporate world.

Establishing operating procedures and governance models are the fundamental concepts of the methodology proposed in this article. The presented use case demonstrates that the proposed steps help companies implement IPA successfully.

**Author Contributions:** Conceptualization, F.A.L.-M., J.D.F.-L. and J.W.B.-B.; methodology, F.A.L.-M. and J.D.F.-L.; validation, F.A.L.-M. and J.D.F.-L.; formal analysis, F.A.L.-M., J.D.F.-L., D.B. and J.W.B.-B.; resources, F.A.L.-M.; writing—original draft preparation, F.A.L.-M., D.B. and J.A.J.-B.; writing—review and editing, F.A.L.-M. and J.A.J.-B.; supervision, J.D.F.-L. and D.B.; project administration, J.D.F.-L. All authors have read and agreed to the published version of the manuscript.

**Funding:** This research received no external funding.

**Informed Consent Statement:** Not applicable.

**Conflicts of Interest:** The authors declare no conflict of interest.

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
