# Peer review of "Intelligent Process Automation: An Application in Manufacturing Industry"

_sustainability, doi:10.3390/su14148804_

Round 1

Reviewer 1 Report

The IEEE Standards Association defines Intelligent Process Automation (IPA) as a preconfigured software instance that combines business rules, experience-based context determination logic, and decision criteria to initiate and execute multiple interrelated human and automated processes in a dynamic context [18].

Sadly, reviewed manuscript entitled "Intelligent Process Automation: An Application in Manufacturing Industry" no contributes to this field of knowledge. The presented material can be a part for a report at industry conference, but not an article in a highly ranked journal like "Sustainability".

Authors are confused about what the article focus. For example, in the introduction (line 51 to 53) they claim that "the primary purpose of this article is to lay out a new and novel methodology that walks through artificial intelligence incorporation into robotic process automation". But in the conclusion section (line 424 to 426), authors state "RPA and AI are two very different technologies that complement each other very well. This paper shows how these technologies can be combined and a methodology for building and successfully deploying IPAs". It is inadequate to base any critical judgement without any specific results supporting this judgement.

This manuscript is a mix of 50 links. "Quintessence" can be a single figure on page 9. Why is this scanned image of terrible quality here?

Above shortcomings are serious ones. Reviewed manuscript not provide an increment of scientific knowledge.

Reviewer 2 Report

The paper presents a new and novel methodology that walks through artificial intelligence incorporation into robotic process automation.

The paper is structured in the first part by reviewing RPA and IPA topics. In contrast, the second presents an application case. It is not clear how the conclusions drawn from the first part are reused in the application case.

The abstract must be wholly restructured following the instructions given here: https://www.mdpi.com/journal/sustainability/instructions

Although the authors indicate the purpose of the paper in the introduction, its original contribution is not evident. The novelties are not clear. State of the art should be further discussed and compared with the paper's objective. The authors do not report any references to papers published in Sustainability. They should consider whether this journal is the best editorial location for the proposed work.

The Materials and Methods section looks more like an extension of state of the art than a research methodology presentation. The Intelligent Process Automation section provides the research background. Authors are advised to review the first part of the paper and make it more consistent with a scientific paper.

In chapter 4, the authors should provide more information about how the review was conducted (e.g., research questions, amount of articles searched, keywords, their analysis, etc.).

The application in an industrial case study is presented quickly. There are no links between this application, methodology, and results previously obtained. The application is interesting for providing the benefits related to IPA implementation.

Reviewer 3 Report

The paper shows essential concepts to create Intelligent Process Automation (IPA) in industries. The main contribution of the paper is the proposal of a framework to implement IPA technologies successfully.

The paper is generally legible and technically good, but it needs to be much better organized. The following issues should be addressed to improve the quality of the paper:

  1. In Section 1 – “Introduction”, the authors write about the need for Digital Transformations (DT) in business processes. Further, they show the basic possibilities of Robotic Process Automation (RPA) and explain how these possibilities can be increased by combining RPA and Artificial Intelligence (AI). Finally, at the end of Section 1, the primary purpose of this paper is defined as “a new methodology that walks through AI incorporation RPA” (lines 51-53). However, after that, in Section 4 – “Methodology” (it seems that the title of the Section is not in accordance with its content), i.e. in Section 4.1 – “The Implementation of Digital Technologies in the Industries”, the authors write once again about the need for DT as well as the fact that there is no method or clear and organized plan for DT implementation that enables business model transformation. Finally, at the end of Section 4.1, the primary purpose of this paper is defined as “addressing of this gap, seeking to offer crucial contributions towards implementing successful Intelligent Automation projects across organizations and demonstrating the importance of this DT” (lines 211-213). It is certainly in accordance with the previously defined. Therefore, I propose a complete revision of Section 1 in such a way that Section 4.1 will combine with Section 1 into a new Section 1 – “Introduction”. At the end of this new Section 1, it is necessary to define clearly the main goal of the paper. Also, for easier understanding, it is desirable to give a brief overview of the paper (for example, one short paragraph) at the end of Section 1.
  2. Section 2 is entitled “Materials and Methods”. It seems that the title of the Section is not in accordance with its content at all. Namely, this Section shows RPA in detail, as well as its advantages and disadvantages. I propose changing the title (for example, “The basics of RPA” or similar).
  3. Section 3 is entitled “Intelligent Process Automation” and gives a good explanation of what IPA is and how it is used. Table 1 shows a comparison between RPA and IPA. Further, I think that the main goal of the paper is highlighted (this should also be mentioned in the new introduction) at the end of this Section (lines 167-168) as “steps which are needed to ensure successful IPA project implementations”. However, after that, in Section 4 – “Methodology”, i.e. in Section 4.2 – “Intelligent Process Automation in the Literature and Consulting Websites”, a review of the literature is given. Therefore, I propose a revision of Section 3 in such a way that Section 4.2 will combine with Section 3 into a new Section 3 – “Intelligent Process Automation” as well as Section 3.1-“The state of art”. Finally, at the end of this new Section 3, the main goal of the paper must be highlighted as “steps which are needed to ensure successful IPA project implementations”.
  4. As parts of the old Section 4 should be integrated into Section 1 and Section 3, the new Section 4 would be “Framework to Implement IPA in Industries” (without a set of words “Results and discussion"). Also, I propose that the steps for IPA implementation be presented in the form of an algorithm at the beginning of this section, in order to make it easier to follow the later text. It could also be a graphic abstract of the paper.
  5. The ordinal numbers of the old Section 6 and Section 7 should be changed further. Now it would be the new Section 5 and Section 6. A set of words “Results and discussion” is something I would include in the title of the new Section 5 – “Results and discussion: Application in the Manufacturing Industry” because that section shows the key results (one example of IPA implementation) which is the basis for the most important conclusions.
  6. Unfortunately, I could not find references under ordinal numbers 6, 15, 33, 36, 49 and 50. I would ask the authors to just check again. Some references seem to have been deleted and some may have an incomplete link.

Round 2

Reviewer 1 Report

Regarding authors reply.

Dear authors declare: «According to the journal scope, it was identified that the presented paper has contributed to the Innovation of Applied systems. The paper contributes to the field of knowledge according to the topics of the journal».

Let's go to the link:

https://www.mdpi.com/topics/app_system

The main goal of this topic is to discover new scientific knowledge relevant to IT-based intelligent mechanical systems, mechanics and design innovations, and applied materials in nanosciences and nanotechnology.

And also I suggest to follow the link:

https://www.mdpi.com/journal/sustainability/instructions#suppmaterials

The journal considers all original research manuscripts provided that the work reports scientifically sound experiments and provides a substantial amount of new information.

My opinion as a reviewer: the second version has not undergone any significant changes. Presented materials are uninformative. Unfortunately, reviewed manuscript entitled "Intelligent Process Automation: An Application in Manufacturing Industry" no contributes to this field of knowledge.

Author Response

Author´s reply to the review report

Reviewer 2 Report

The authors, overall, have provided a reply to the reviewer’s comments that sometimes were not enough. Thus, the reviewer suggests the following remarks.

The abstract is still too short for presenting the paper. Further information concerning methods and results should be provided.

The paper’s novelties still need to be clearly highlighted.

The authors inserted the research questions concerning the scientific review. How do the identified articles relate to these questions? The authors should argue more about how Table 2 was defined from the search query.

At the beginning of chapter 6 it would be helpful to insert a short paragraph connecting with the methodology (Figure 1).

Author Response

Author´s reply to the review report

Reviewer 3 Report

Almost all of my comments were taken into account. All the questions are well answered, and the article can be published.

Author Response

Author´s reply to the review report

Round 3

Reviewer 1 Report

The manuscript has been sufficiently improved. Overall, this work is ready for publication in present form.

Reviewer 2 Report

The authors have addressed all the reviewer’s comments. The paper can be accepted in its current form.